# The challenge of chromatin model comparison and validation: A project from the first international 4D Nucleome Hackathon

Jędrzej Kubica[1,2‡], Sevastianos Korsak[1,3‡], Krzysztof H. Banecki[1,3], Dvir Schirman[4], Anurupa Devi Yadavalli[5], Ariana Brenner Clerkin[6,7], David Kouřil[8], Michał Kadlof[3], Ben Busby[9], Dariusz Plewczynski[1,3]*

**1** Laboratory of Functional and Structural Genomics, Centre of New Technologies, University of Warsaw, Warsaw, Poland, **2** Univ. Grenoble Alpes, CNRS, UMR, TIMC/ MAGe, Grenoble, France, **3** Faculty of Mathematics and Information Science, Warsaw University of Technology, Warsaw, Poland, **4** Department of Cell and Molecular Biology, Uppsala University, Uppsala, Sweden, **5** Department of Immunobiology, School of Medicine, Yale University, New Haven, Connecticut, United States of America, **6** Laboratory of Genome Architecture and Dynamics, The Rockefeller University, New York, New York, United States of America, **7** Tri-Institutional PhD Program in Computational Biology and Medicine, New York, New York, United States of America, **8** Department of Biomedical Informatics, Harvard Medical School, Boston, Maryland, United States of America, **9** DNAnexus, Mountain View, California, United States of America

‡ These authors should be regarded as co-first authors.
* Dariusz.Plewczynski@pw.edu.pl, d.plewczynski@cent.uw.edu.pl

## Abstract

The computational modeling of chromatin structure is highly complex due to the hierarchical organization of chromatin, which reflects its diverse biophysical principles, as well as inherent dynamism, which underlies its complexity. Chromatin structure modeling can be based on diverse approaches and assumptions, making it essential to determine how different methods influence the modeling outcomes. We conducted a project at the NIH-funded 4D Nucleome Hackathon on March 18–21, 2024, at The University of Washington in Seattle, USA. The hackathon provided an amazing opportunity to gather an international, multi-institutional and unbiased group of experts to discuss, understand and undertake the challenges of chromatin model comparison and validation. Here we give an overview of the current state of the 3D chromatin field and discuss our efforts to run and validate the models. We used distance matrices to represent chromatin models and we calculated Spearman correlation coefficients to estimate differences between models, as well as between models and experimental data. In addition, we discuss challenges in chromatin structure modeling that include: 1) different aspects of chromatin biophysics and scales complicate model comparisons, 2) large diversity of experimental data (e.g., population-based, single-cell, protein-specific) that differ in mathematical properties, heatmap smoothness, noise and resolutions complicates model validation, 3) expertise in biology, bioinformatics, and physics is necessary to conduct comprehensive research on chromatin structure, 4) bioinformatic software, which is often developed

**Data availability statement:** To ensure that these results are reproducible, all scripts for model comparison and validation have been made publicly available on GitHub: https://github.com/SFGLab/Polymer_model_benchmark.

**Funding:** DP, JK, MK, SK, KB research was funded by Warsaw University of Technology within the Excellence Initiative: Research University (IDUB) programme, their work has been co-supported by Polish National Science Centre (Narodowe Centrum Nauki) (2020/37/B/NZ2/03757) and the National Institute of Health USA 4DNucleome grant 1U54DK107967-01 "Nucleome Positioning System for Spatiotemporal Genome Organization and Regulation". DK was supported in part by the National Institutes of Health (R01HG011773 and UM1HG011536). The funders had no role in study design, data collection and analysis, decision to publish, or preparation of the manuscript.

**Competing interests:** I have read the journal's policy and the authors of this manuscript have the following competing interests: BB is a full-time employee of DNAnexus.

in academic settings, is characterized by insufficient support and documentation. We also emphasize the importance of establishing guidelines for software development and standardization.

## Author summary

Current computational methods for chromatin modeling consider different chromatin biophysics, scales and assumptions, which complicate software comparison. In this work, we provide an overview of state-of-the-art software for chromatin structure modeling, discuss the challenges of chromatin model comparison and validation, and discuss the difficulties with running the software and interpreting the results. To address those challenges, we gathered a diverse and unbiased group of experts at the 4D Nucleome Consortium Hackathon on March 18–21, 2024, at The University of Washington in Seattle, USA. During the hackathon, we developed a bioinformatic workflow for chromatin model comparison and validation that provides a future reference for researchers in the field. We believe that our results will benefit the future development of software for chromatin structure modeling. Furthermore, we emphasize the importance of establishing guidelines for software development and standardization that would have a long-term impact on the 3D genomics community.

## Introduction

### Hierarchical organization of chromatin structure

Modeling 3D chromatin structure requires an examination of its multi-scale organization (Fig 1). At the fundamental level, DNA wraps approximately 1.65 times around an octamer of histone proteins to form nucleosomes [1,2]. These nucleosomes play pivotal roles in templating many biophysical processes through histone modifications [3]. Despite previous conjectures regarding the formation of a 30 nm fiber through nucleosome clustering, contemporary consensus does not support this 30 nm modeling *in vivo* [3–8]. The intermediate scale encompasses loops and topologically associated domains (TADs), predominantly governed by two principal proteins: structural maintenance of chromosomes (SMC) complexes [9], characterized by ring-like configurations facilitating loop extrusion within chromatin, and CCCTC-binding factor (CTCF) loops [10], exhibiting prolonged lifespans while binding to specific motifs. Consequently, SMC complexes assume a ring-like conformation to extrude loops, while CTCF functions as orientation-dependent impediments to the former. At the sub-megabase level, the communication between these loops is restricted by the topologically associated domains, enabling cell type-specific gene expression programs. At the compartment level, chromatin segregates into two compartments: open (A), which is loosely arranged and biophysically accessible to transcription factors, and closed (B), which is denser [11]. Typically, compartmentalization is modeled

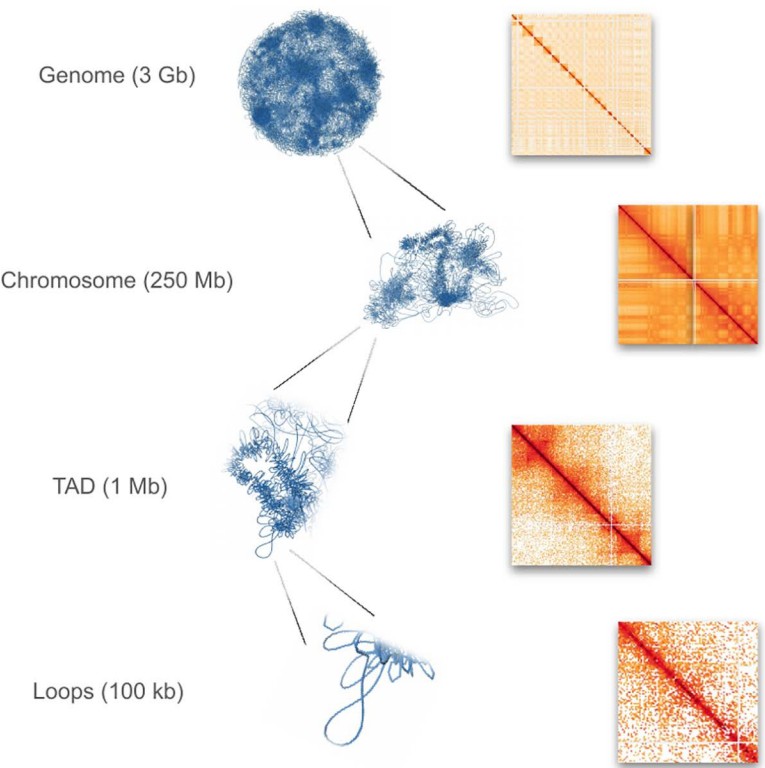

**Multi-scale chromatin organization**

Genome (3 Gb)

Chromosome (250 Mb)

TAD (1 Mb)

Loops (100 kb)

**Fig 1. An illustration of the multi-scale chromatin organization.** The approximate genome lengths (in base pairs) are presented alongside structural models and experimental contact maps for each scale (data: 4DN Data Portal 4DNES4AABNEZ - in situ Hi-C on human embryonic stem cells (H1) treated with RNase A). Abbreviation: TAD - topologically associated domain.

employing long-range non-bonding forces, such as block-copolymer potentials [12], representing an amalgamation of diverse smaller-scale interactions. Finally, the human genomic landscape is demarcated into 23 pairs of chromosomes, each resembling an independent polymer chain [13]. These chromosomes are densely packed in the nucleus, with specific regions (lamina-associated domains) exhibiting proximity to the nuclear lamina [13,14].

It appears that each scale of the chromatin organization functions as an autonomous biological apparatus. Intriguingly, despite this autonomy, inter-scale interactions have been observed [15,16]. For instance, histone modifications are purportedly instrumental in shaping regions of open and closed chromatin, while compartmentalization correlates with lamina domains, evidenced by the propensity of B compartments to interact with the lamina [17]. In conclusion, the intricate interplay of diverse proteins dynamically interacting with chromatin, combined with the complexity at each level of organization, underscores the multifaceted nature of chromatin structure.

Various experimental methodologies have been devised to probe each scale individually. For instance, MNase-seq and ATAC-seq data [18,19] determine the nucleosome positioning. ChIA-PET and HiChIP experiments [20,21] produce contact matrices and prove efficacious in loop-scale modeling, while Hi-C experiments [16] identify compartments and subcompartments, thus, an integration thereof can be useful in chromatin modeling. Compared to Hi-C data, which has been extensively used for 3D chromatin modeling [22,23], methods such as ChIA-PET and HiChIP have received significantly less attention for reconstructing full 3D chromatin structures, since they detect interactions mediated by specific proteins

or histone modifications. As a result, these techniques are often considered less suitable for global chromatin conformation analysis. However, some computational models described in our study, specifically those that aim to explicitly model the loop extrusion process, have been designed to accommodate these data types, prompting us to include ChIA-PET data in our hackathon project. Since these methods primarily highlight interactions associated with specific histone marks or proteins, their interpretation requires careful consideration.

Beyond Hi-C, ChIA-PET, and HiChIP, other chromatin conformation capture techniques avoid the limitation of protein specificity (for a review, see [24]). A particularly noteworthy example is Micro-C [25] and its improved variant, Micro-C XL [26]. Those methodologies are known for providing high-resolution contact maps due to their use of micrococcal nuclease digestion, which can be beneficial for studying fine-scale chromatin architecture, especially on a nucleosome level. While Micro-C has not yet been widely applied for 3D chromatin structure reconstruction, at least not to the same extent as Hi-C and related techniques, it provides valuable insights into chromatin organization at the nucleosome level. Due to its high-resolution nature, Micro-C has been employed in some studies as an independent validation tool for computational chromatin models [27–29], demonstrating its potential as a complementary data source for chromatin conformation analysis.

Nonetheless, a critical consideration often overlooked is the population and cell cycle averaging inherent in many of these datasets, necessitating the adoption of single-cell experimental techniques such as single-cell Hi-C (scHi-C) to mitigate this limitation [30]. Although many experimental methods were developed, there remains a lack of sufficient data that hinders a full understanding of chromatin structure. This data gap presents a new challenge: comparison and validation of 3D modeling techniques.

## Modeling of the chromatin structure

Theoretical modeling of the chromatin structure is highly complex. It requires consideration of various factors that influence the final configuration of the polymer, as well as its multi-scale organization. At the lowest level, short-range interactions between residue pairs are predominant. However, at higher levels, weaker long-range interactions maintain the compactness of the polymer in the nucleus. Incorporating more biological information models is more realistic, however, computational efficiency constitutes a significant challenge. In addition, chromatin modeling faces other obstacles, such as a lack of method standardization and evaluation metrics, proper model visualization, and dealing with experimental data averaged over time and population. To address them, various approaches have been developed in recent years to model chromatin structure at different scales and resolutions. Despite these efforts, a gold standard modelling approach has not been established, primarily because model validation against experimental data remains difficult.

## Methods for chromatin structure modeling

Strategies for chromatin structure modeling can be divided into data-driven and predictive (Fig 2). The data-driven strategies take as input experimental genomic data (e.g., Hi-C or ChIA-PET that provide contact frequencies) or imaging data (e.g., FISH that shows polymer density). Predictive strategies, propelled by advancements in deep learning, analyze data from ChIP-seq, ATAC-seq, or DNA sequencing experiments, encompassing epigenetic modifications or chromatin accessibility, to infer chromatin structure [31,32]. Output may include a contact map, a 3D model or an ensemble of 3D models, categorized by the scale of modeling, such as loops, TADs, or the whole genome. Input data can be derived from "bulk" or single-cell experiments. Constructing models from "bulk" Hi-C data presents challenges due to averaging chromatin states, overlooking the intrinsic heterogeneity of the underlying chromatin conformation changes. To alleviate those problems, a slightly different set of methods was developed that model the chromatin conformation in particular cell-based states based on scHi-C data first introduced by Nagano et al., 2013 [30]. Those methods were meant to deal with the specific focus on the sparsity of the input data, which is the main problem of scHi-C. Several methods, which are robust to the data sparsity, have already been implemented [33–43] and were briefly reviewed before [44]. Most of them use scoring functions that are optimized by simulated annealing protocols or gradient descent optimization [38,41]. Other

methods define the posterior Bayesian probability function and apply Markov Chain Monte Carlo (MCMC) algorithms to draw models from the distribution [37,40] or opt for molecular dynamics simulations [36]. These methods are written in C++ or Python and are easily accessible and usable.

Single-cell multi-omics approaches are designed to simultaneously capture various chromatin data modalities at the single-cell level. These methods aim at the concurrent acquisition of chromatin interactions and gene expression data, typically integrating Hi-C and RNA sequencing (RNA-seq). Examples of such multi-omics methods include scCARE-seq [45], HiRES [46], GAGE-seq [47], and LiMCA [48]. Other approaches simultaneously capture the chromatin interactions and methylation (e.g., sn-m3C-seq [49] and Methyl-Hi-C [50]). Furthermore, integrative multi-omic methods are currently under active development. A recent example is ChAIR [51], which enables the joint profiling of chromatin accessibility, chromatin interactions, and gene expression at the single-cell resolution. Although some methods for chromatin reconstruction leveraging supplementary data in addition to Hi-C have been proposed [52], relatively few approaches have been specifically designed for this purpose. Given the rapid advancements in the field, a significant expansion of such methodologies is expected in the coming years.

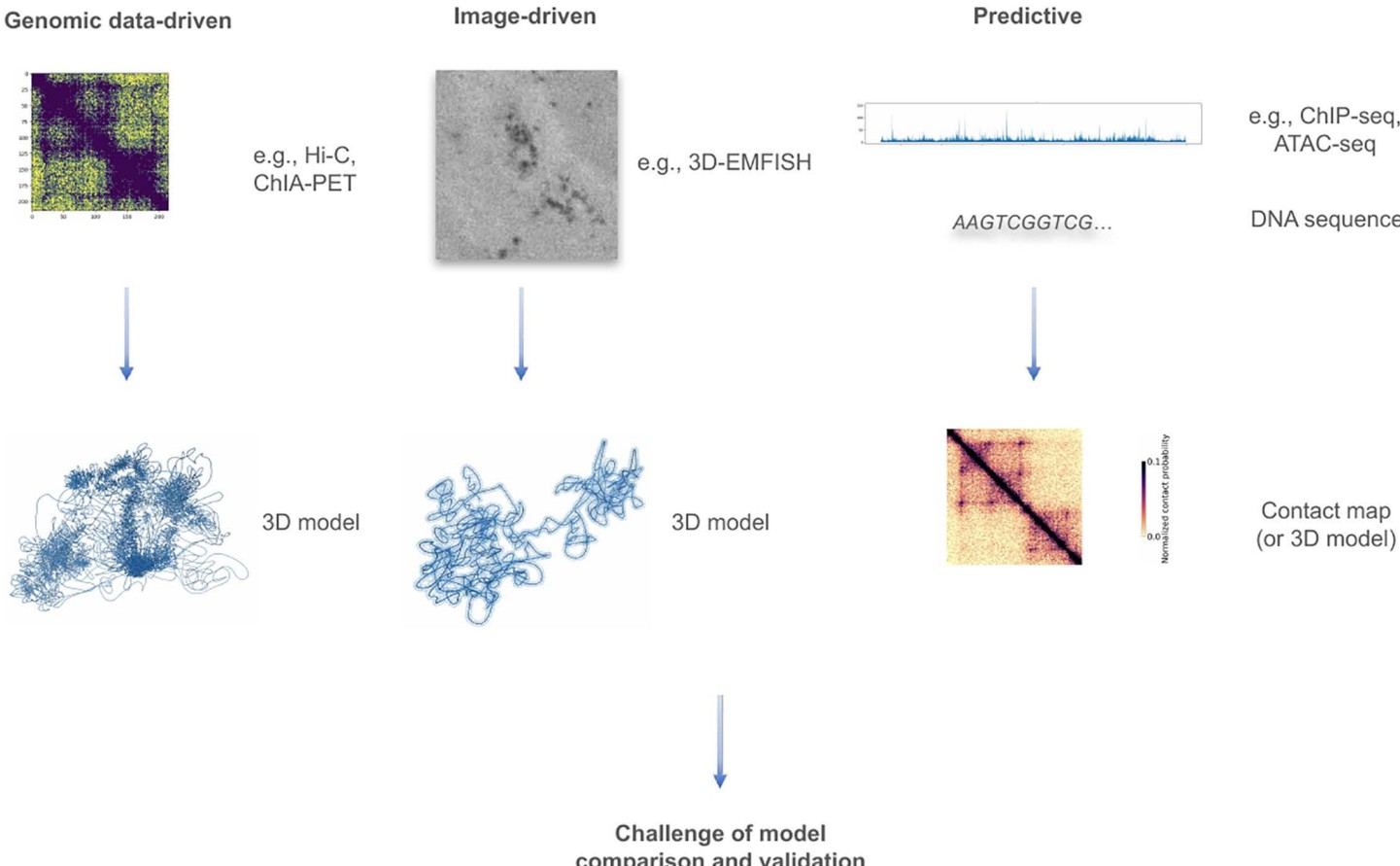

**Fig 2. An overview of the strategies for chromatin structure modeling.** Strategies for chromatin modeling can be divided into genomic data-driven, image-driven and predictive. Each strategy can produce a structural 3D model or a contact map.

Another set of modeling challenges arises from methods designed to capture multi-way chromatin interactions. Notable examples include ChIA-Drop [53], scSPRITE [54], GAM [55], and immunoGAM [56]. These sequencing-based, ligation-free approaches offer an alternative means of investigating 3D chromatin conformation and may help address challenges associated with single-cell data sparsity. However, modeling approaches for analyzing such data require distinct computational strategies.

## Bottom-up versus top-down modeling

Primarily, bottom-up approaches employ the first principal assumptions regarding the system's force field to reconstruct the chromatin conformation (e.g., Spring model, MultiMM) [57,58]. These models incorporate loops derived from Hi-C or ChIA-PET data, utilizing virtual springs to bring spatially distant chromatin regions into proximity. Conversely, top-down models (e.g., MiChroM, GEM-reconstruction, PHi-C, miniMDS) [59–62] prioritize the optimization of model hyperparameters to emulate experimental Hi-C heatmaps, ensuring strong correlation between all-versus-all distances of the polymer and Hi-C data, rather than emphasizing the force field. Moreover, stochastic models (e.g., MoDLE, LoopSage) [63,64] generate thermodynamic ensembles of models, wherein the average all-versus-all distances replicate TAD structures. These models, characterized by escalating complexity, endeavor to reconstruct heatmaps by incorporating biophysical assumptions regarding loop extrusion and modeling dynamic trajectories over time. Minimalistic data, such as anchors from Hi-C, ChIA-PET, or HiChIP experiments, or information gleaned from ChIP-seq experiments, are utilized to infer the locations and orientations of barrier CTCF proteins. Therefore, successful modeling necessitates the adjustment of numerous biophysical parameters about loop extrusion dynamics.

Below, we present a general overview of the variety of approaches in the software for chromatin structure modeling (Tables 1 and 2), which were also described in detail in recent literature on this topic [22,65–67]. Most of the methods discussed in this study can be broadly categorized as genomic data-driven models, with the majority designed for bulk Hi-C data and some specifically tailored for single-cell data. Additionally, Table 2 presents several other methods that were initially considered for inclusion in our study but were ultimately excluded, as they belong to a different category of chromatin modeling approaches. These excluded methods often focus on image-driven modeling or the estimation of Hi-C contact maps, rather than generating 3D polymer models.

## 4D Nucleome Hackathon 2024

Hackathons are popular in life sciences, especially in the field of genomics, because they offer an amazing opportunity to foster international multi-disciplinary collaboration and to quickly advance projects based on the principle of open innovation [102,103]. The advantage of doing software comparison in a hackathon setting is that one can collect diverse, unbiased, yet expert views on the software. Following this emerging idea, we participated in the 4D Nucleome Hackathon 2024, organized by the 4D Nucleome Consortium, that took place on March 18–21, 2024, at The University of Washington in Seattle, USA (event website: https://hack4dnucleome.github.io/). One of the 4D Nucleome Consortium aims is to study the structure and function of the human genome through predictive models of chromatin [104,105], however, currently, criteria for comparison and validation of such models are lacking, even though there have been initiatives undertaken to benchmark computational methods for chromatin modeling [65,66,106]. The hackathon offered an opportunity to review the current state of the 3D chromatin modelling field, as well as to define criteria for chromatin model comparison and validation. Although defining such criteria might seem conceptually straightforward (based on principles from polymer physics and statistical modeling), their implementation is highly complex and challenging due to the variability in experimental data and computational approaches. The objective of this hackathon project was to investigate different methodologies for chromatin modelling. It involved running five software packages, and comparing their outputs to each other and experimental data. By testing different models, we aimed at identifying their strengths and limitations, as well as highlighting key challenges in model comparison and validation.

**Table 1. Overview of software for chromatin structure modeling for genomic data-driven strategy. Software for chromatin modeling can be categorized into different categories based on input data, modeling scale or output format.**

| Software name | URL | Reference | Input data | Scale | Strategy | Output |
|---|---|---|---|---|---|---|
| Nuc-dynamics | https://github.com/tjs23/nuc_dynamics | [36] | Hi-C | TAD, chromosome, genome | Genomic data-driven (single-cell) | 3D model |
| SCL | http://dna.cs.miami.edu/SCL/ | [37] | Hi-C | TAD, chromosome, genome | Genomic data-driven (single-cell) | 3D model |
| LoopSage | https://github.com/SFGLab/LoopSage | [64] | ChIA-PET, Hi-ChIP, ChIP-Seq | loops | Genomic data-driven | 3D model |
| 3D-Gnome 3.0 | https://3dgnome.mini.pw.edu.pl/ | [68] | ChIA-PET, VCF | chromosome | Genomic data-driven | 3D model |
| MoDLE | https://github.com/paulsengroup/modle | [63] | ChIP-seq ATAC-seq | loops | Genomic data-driven | 3D model |
| HiP-HoP | | [69] | ChIP-seq ATAC-seq, DNA-seq | loops, chromosome | Genomic data-driven | 3D model |
| InfMod3DGen | https://github.com/wangsy11/InfMod3DGen | [70] | Hi-C | chromosome | Genomic data-driven | 3D model |
| BACH | http://www.fas.harvard.edu/~junliu/BACH/ | [71] | Hi-C | chromosome | Genomic data-driven | 3D model |
| ChromSDE | | [72] | Hi-C | chromosome | Genomic data-driven | 3D model |
| ShRec3D+ | | [73] | Hi-C | chromosome | Genomic data-driven | 3D model |
| LJ3D | http://dna.cs.miami.edu/LJ3D/ | [40] | Hi-C | chromosome | Genomic data-driven | 3D model |
| HSA | http://ouyanglab.jax.org/hsa/ | [74] | Hi-C | chromosome, genome | Genomic data-driven | 3D model |
| MiChroM | https://open-michrom.readthedocs.io/en/latest/OpenMiChroM.html | [75] | Hi-C | chromosome, genome | Genomic data-driven | 3D model, contact map |
| GEM reconstruction | https://github.com/gletreut/gem_reconstruction | [60] | Hi-C | chromosome, genome | Genomic data-driven | contact map |
| PGS | | [76] | Hi-C | genome | Genomic data-driven | 3D model |
| LMS algorithm | | [77] | Hi-C | loops | Genomic data-driven | 3D model |
| MDS | | [78] | Hi-C | loops, TAD, chromosome, genome | Genomic data-driven | 3D model |
| TADbit | https://github.com/3DGenomes/TADbit | [79] | Hi-C | TAD | Genomic data-driven | 3D model |
| AutoChrom3D | http://ibi.hzau.edu.cn/3dmodel/ | [80] | Hi-C | TAD | Genomic data-driven | 3D model |
| HiCNet | http://dna.cs.miami.edu/HiCNet/ | [81] | Hi-C | TAD, chromosome | Genomic data-driven | 3D model |
| miniMDS | https://github.com/seqcode/miniMDS | [62] | Hi-C | TAD, chromosome, genome | Genomic data-driven | 3D model |
| MOGEN | https://github.com/BDM-Lab/MOGEN | [82] | Hi-C | TAD, chromosome, genome | Genomic data-driven | 3D model |
| MCMC5C | http://dostielab.biochem.mcgill.ca/ | [83] | Hi-C, 5C | chromosome | Genomic data-driven | 3D model |
| looper | https://github.com/plewczynski/looper | [84] | Hi-C, ChIA-PET | loops | Genomic data-driven | 3D model, contact map |
| Spring-model | https://spring-model.mini.pw.edu.pl/ | [57] | Hi-C, ChIA-PET | TAD | Genomic data-driven | 3D model |
| PHi-C | https://github.com/soyashinkai/PHi-C | [61] | Hi-C, ChIA-PET | TAD, chromosome | Genomic data-driven | 3D model |
| PHi-C2 | https://github.com/soyashinkai/PHi-C2 | [85] | Hi-C, ChIA-PET | TAD, chromosome | Genomic data-driven | 3D model |
| HIPPS-DIMES | https://github.com/anyuzx/HIPPS-DIMES | [86,87] | Hi-C, ChIA-PET | TAD, chromosome, genome | Genomic data-driven | 3D model |
| Chrom3D | https://github.com/Chrom3D/Chrom3D | [88] | Hi-C, TCC, 5C, ChIP-seq | TAD, chromosome. genome | Genomic data-driven | 3D model |
| MultiMM | https://github.com/SFGLab/MultiMM | [58] | Hi-C, Hi-ChIP, ChIA-PET | loops, TAD, chromosome, genome | Genomic data-driven | 3D model |

**Table 2. Examples of software for chromatin structure modeling made with other than genomic data-driven strategy. Software for chromatin modeling can be categorized into different categories based on input data, modeling scale, modeling strategy or output format.**

| Software name | URL | Reference | Input data | Scale | Strategy | Output |
|---|---|---|---|---|---|---|
| Chromatin states-based model | https://github.com/ZhangGroup-MITChemistry/DRAGON | [89] | ChIP-seq | loops, TAD, chromosome | Predictive - physics based | 3D model |
| C.Origami | https://github.com/tanjimin/C.Origami | [90] | DNA sequence, ChIP-seq, ATAC-seq | chromosome | Predictive - deep learning | contact map |
| HiCdiffusion | https://github.com/SFGLab/HiCDiffusion | [91] | DNA sequence | chromosome | Predictive - deep learning | contact map |
| Akita | https://github.com/calico/basenji/tree/master/manuscripts/akita | [92] | DNA sequence | chromosome, genome | Predictive - deep learning | contact map |
| DeepC | https://github.com/rschwess/deepC | [93] | DNA sequence | TAD, chromosome, genome | Predictive - deep learning | contact map |
| ChromFormer | https://github.com/AI4SCR/ChromFormer | [31] | Hi-C | loops, TAD | Predictive - deep learning | 3D model, contact map |
| TECH-3D | https://github.com/AI4SCR/tech3d | [94] | Hi-C | TAD, chromosome, genome | Predictive - deep learning | 3D model |
| REACH-3D | | [95] | Hi-C | TAD, chromosome, genome | Predictive - deep learning | 3D model |
| DHL | | [96] | DNA sequence | loops | Predictive - deep hybrid learning | contact map |
| ORCA | https://github.com/BoettigerLab/ORCA-public | [97] | image | TAD | Image-driven | 3D model |
| Md-soft, MDFF | | [98] | PDB, image | loops, TAD, chromosome | Genomic data and image-driven | 3D model |
| GEM-FISH | https://github.com/ahmedabbas81/GEM-FISH | [99] | Hi-C, FISH | TAD, chromosome | Genomic data and image-driven | 3D model |
| Martini | http://cgmartini.nl/index.php/tutorials-general-introduction-gmx5/tutorial-martini-dna-gmx5 | [100] | DNA sequence | all-atom, coarse-grained structure | All-atom, Coarse-grained | 3D model |
| CollepardoLab_Chromatin_Model | https://github.com/CollepardoLab/CollepardoLab_Chromatin_Model | [101] | DNA sequence | all-atom, coarse-grained structure | All-atom, Coarse-grained | 3D model |

During the hackathon, due to a limited time frame and resources, we focused our efforts on five distinct software packages (DIMES, MultiMM, MiChroM, LoopSage, and PHi-C2), which allowed us to demonstrate the underlying challenges of chromatin structure modeling. We selected methods to be used in our study based on the following criteria:

1. The method is published in a peer-reviewed journal or as a preprint on https://www.biorxiv.org/ with evidence of a good performance.

2. The method is implemented as a well-documented software package distributed under a public license (MIT or GNU GPL v2.0/3.0).

3. The method is designed to model chromatin structure in the loop and/or TAD resolution.

4. The method is directly applicable to Hi-C and/or ChIA-PET data without additional input data.

We selected five methods from Table 1 that met all the necessary criteria. Rather than conducting a direct comparison or ranking of the software, our focus was to explore specific aspects of different chromatin modeling strategies. Our analyses

produced qualitatively different results, depending on the modeling strategy—for example, ensemble modeling based on loop extrusion in LoopSage, versus ensemble modeling using contact probability inference from population-averaged data in HIPPS-DIMES.

Although this study is not intended to be a comprehensive review of all existing 3D chromatin structure reconstruction methodologies, the analysis of five methods allowed us to draw meaningful conclusions about the state-of-the-art and discuss method comparison and validation.

## Results

### Challenge of benchmarking

The main objective of this project was to address the challenges of chromatin model vs. model comparison, as well as the validation of chromatin models using experimental data (Hi-C [107], ChIA-PET [20] and SPRITE [108]). Our hackathon experience highlighted several key challenges associated with benchmarking of chromatin models. These challenges arise from multiple factors: 1) bioinformatic software is frequently developed in academic settings and often lacks long-term support, resulting in many outdated software with insufficient support and poor documentation [109]; 2) various models are designed to address different aspects of chromatin biophysics, focusing on diverse biophysical problems or scales, complicating direct comparisons; 3) the complexity of chromatin folding research necessitates expertise in biology, bioinformatics, and physics, which hinders the development of simple and user-friendly models. These challenges are due to variations in implementations or differences in the underlying biophysical principles they capture. For instance, sampling techniques like Metropolis or Simulated Annealing encompass a diverse set of methods (replica-exchange variables, or generalization of simulated annealing approach), each defined by distinct hyperparameters (e.g., temperature or sampling frequency) that can significantly influence the model [110]. Despite these challenges, we successfully ran several modeling methods and developed a workflow for their benchmarking (Fig 3).

We began the project by executing each software listed in Table 1 consecutively and independently. If we encountered significant difficulties with a particular software, we opted to move on to the next one, bearing in mind the limited time-frame of the hackathon (4 days). The chromatin modeling software produced 3D models, which we visualized to examine the differences between them. Then we converted the models into 2D contact matrices, which allowed us to calculate correlation coefficients. We assigned a numerical value to each pair of matrices that represented the differences between the models. We designed an easily programmable way to convert the models into matrices, and once we collected the matrices, we chose the Spearman correlation coefficient for matrix comparison. We discussed alternative metrics for comparison between two 3D models (e.g., Pearson correlation coefficient or root mean squared deviation (RMSD)), however, we decided to focus on Spearman correlation because of its simplicity and interpretability.

While the pipeline for chromatin model validation was straightforward to implement in code, accounting for experimental and simulated biases presented a significant challenge. Structural biases, such as the sparsity of heatmaps and the dominant diagonal signal, can lead to unreliable correlation estimates [111]. Additionally, the smoothness of heatmaps varies substantially between experimental and simulated data, further complicating comparisons. Denoising techniques, such as GenomeDISCO [112], can address these issues by estimating connection probabilities using random walks. Another critical challenge is the multi-scale nature of chromatin heatmaps, which can be mitigated by applying distance-stratified metrics. Not all heatmap features contribute equally to structural comparison; compartments, TADs, loops, and stripes are particularly informative, while a significant portion of the heatmap contains less relevant information. A basic approach to distance-stratified metrics involves computing correlations within distance windows, while more advanced methods, such as HiCRep [113], provide single-stratified metrics for structural comparison. However, tools like HiCRep and Genome-DISCO are primarily designed for comparing experimental datasets. When applied to simulation-derived heatmaps, the interpretation is less straightforward, as the relationship between model-derived distances and experimental contact frequencies is not explicitly defined. In Hi-C data, interactions typically follow a power-law decay, $s \sim (1/d)^a$, where "a" must

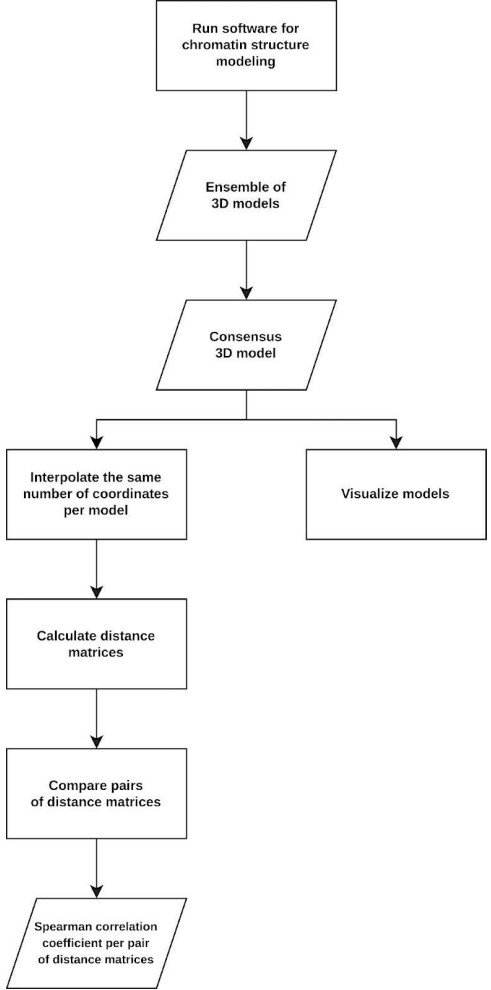

**Fig 3. Workflow for benchmarking of chromatin structural models.** The workflow developed at the 4D Nucleome Hackathon 2024 consists of multiple steps that include obtaining, visualizing, processing and comparing models of chromatin structure.

be determined. A similar relationship may hold for other datasets, such as ChIA-PET, but its validity remains uncertain and requires further investigation. Computer vision metrics, such as the Fréchet Inception Distance (FID) [114,115], which measures the similarity between distributions of high-dimensional data by comparing feature representations extracted from a neural network, could serve as an alternative approach for validating model-derived heatmaps. These metrics provide a more sophisticated way to capture the visual intuition of data patterns. However, they often lack biological interpretability, limiting their applicability in chromatin structure analysis.

## Challenge of model vs. model comparison

During the 4-day hackathon, we followed our workflow to generate and compare chromatin models. We initially hypothesized that the models for the same genomic region would be consistent in shapes and sizes, therefore, we chose a region of approximately 1 megabase (Mb) that was of an appropriate length to model both short- and long-distance interactions. We selected two experimental data types: Hi-C (4DN Data Portal: 4DNES4AABNEZ) and ChIA-PET (CTCF; ENCODE: ENCSR184YZV). We used 2D contact matrices from Hi-C and ChIA-PET as input for five distinct, user-friendly software

packages: DIMES, MultiMM, MiChroM, LoopSage, and PHi-C2. We used Hi-C and ChIA-PET data for the Tier 1 GM12878 human cell line and selected a chunk of it that corresponded to a TAD of approximately 1 Mb (chr1:178.421.513–179.491.193). We obtained output models (in XYZ, PDB, or CIF format) from those five software packages that generated either one model or an ensemble of models, and from each approach, we kept a single representative model. The output models were of different resolutions, therefore, they could not be directly compared. The default resolution for MiChroM, MultiMM, LoopSage and DIMES (Hi-C only) was 1000 base pairs per bead, whereas for PHi-C2 and DIMES (ChIA-PET) it was 5000 base pairs per bead. We first interpolated each model to the same number of coordinates by finding an approximate basis spline representation of it and obtained a uniform resolution across all of them. We settled for a final resolution of 214 beads for each model, equivalent to approximately 5000 base pairs per bead. We did not assess how much information was lost during the interpolation, nor did we pursue the idea of testing other values for resolution due to the limited timeframe of the hackathon. Based on these standardized models, we created distance matrices of consistent shapes. We visualized the output models of the genomic region of interest (chr1:178.421.513–179.491.193) generated using Hi-C data (Fig 4), as well as ChIA-PET data. Despite using the same input data for all software packages, the models displayed inconsistencies in both shapes and sizes. Therefore, we used correlation coefficients, not the model size, as the comparison metric. The reason for the disparities might be due to the differences in modeling methodologies and assumptions used in the software. As previously mentioned, algorithms are designed to address specific aspects of chromatin biophysics and focus on different biological scales, complicating model comparisons. Moreover, we used the default parameters for all software packages, therefore leaving open the possibility that the parameters can influence the modeling process and results.

We hypothesized that using ChIA-PET data, which provides different insights about genome folding compared to Hi-C data, we would obtain different models. This assumption comes from the complementary, but not indistinguishable, nature of both experimental methods. Both can be represented as contact matrices, however, the elements of Hi-C matrices represent contact probabilities between genomic regions, whereas ChIA-PET matrices represent genomic interactions mediated by proteins. We used the same software packages once more, providing ChIA-PET data as input. The output model visualizations demonstrate that all ChIA-PET models differ from each other in shapes (Fig 5). In addition, using ChIA-PET instead of Hi-C does not yield identical structures.

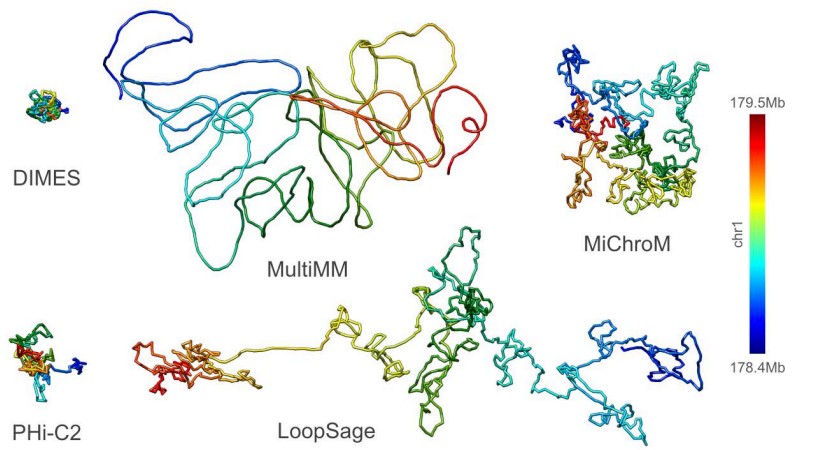

**Fig 4. Chromatin models based on HiC data showing the same topologically associated domain.** Models presenting the same genomic region (chr1:178.421.513–179.491.193) obtained from five software packages (DIMES, MultiMM, MiChroM, LoopSage, and PHi-C2).

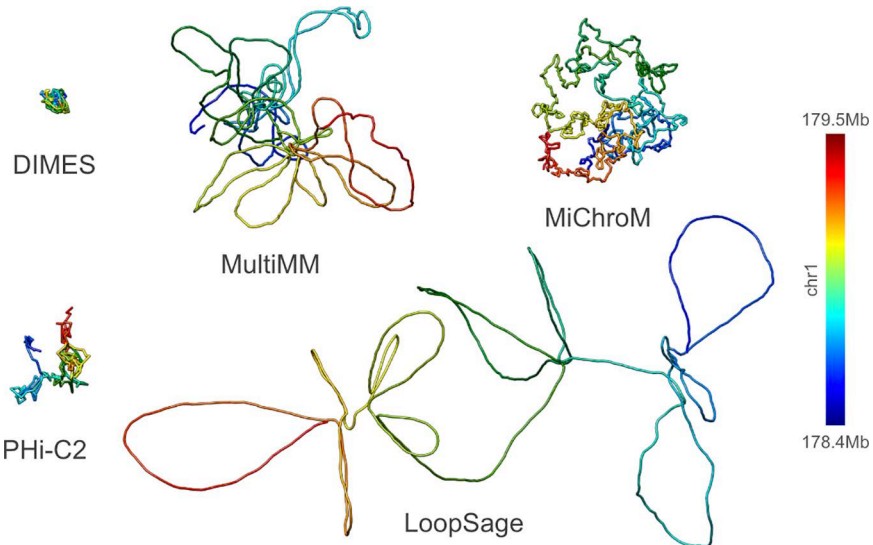

**Fig 5. Chromatin models based on ChIA-PET data showing the same topologically associated domain.** Models presenting the same genomic region (chr1:178.421.513–179.491.193) obtained from five software packages (DIMES, MultiMM, MiChroM, LoopSage, and PHi-C2).

These findings highlight the challenges of chromatin structure modeling and emphasize the need for robust methods to compare models. To address this, we proposed converting the 3D models into distance matrices, allowing us to quantify the differences between them through matrix comparisons. We computed 2D distance matrices for all models, the elements of which corresponded to pairwise Euclidean distances between the beads. Since LoopSage uses simulated annealing, where the ensemble represents a trajectory through decreasing temperatures and only the final structure corresponds to the most stable and physically plausible state, we computed the heatmap from this final structure; in contrast, DIMES generates an ensemble of statistically equivalent structures sampled independently from the same equilibrium distribution, so we averaged the heatmaps over the entire ensemble to reflect the distribution of possible configurations; for non-ensembled methods, we computed the heatmap from a single representative structure. The matrices allowed us to calculate a Spearman correlation coefficient ($\rho$) (that takes a value between -1 and 1) for each pair of matrices to assess similarities between the corresponding models, while the higher the value of the Spearman correlation coefficient, the more similar the models. Other similar metrics (e.g., Pearson correlation coefficient) or combinations thereof might be more appropriate for matrix comparison, however, we selected the most straightforward and intuitive approach for the purpose of the hackathon. We expected to observe rather significant heterogeneity in the coefficients since the models differed notably in shapes, and as anticipated, the coefficients reflected the discrepancies between the models. In terms of the models generated with Hi-C data (Table 3), the highest correlation was observed between the models generated with MultiMM and DIMES ($\rho = 0.725$), while the lowest correlation was observed between the models from PHi-C2 and MiChroM ($\rho = 0.373$). As regards the models generated from ChIA-PET data (Table 4), the highest correlation was found between the models generated with LoopSage and MultiMM ($\rho = 0.800$), while the lowest correlation was shown to be between the models from MiChroM and DIMES ($\rho = 0.248$). Overall, by using Spearman correlation coefficients, we were able to demonstrate a great heterogeneity in chromatin models produced by the tested software packages that were also observed in model visualizations, as well as comparisons of heatmaps. For example, we observed significant differences when we compared heatmaps corresponding to the Hi-C models with the highest and the lowest correlation (Fig 6).

**Table 3. Comparison of chromatin models generated based on Hi-C data. The table contains Spearman correlation coefficients calculated between pairs of distance matrices generated based on the models from five software packages (DIMES, MultiMM, MiChroM, LoopSage, and PHi-C2).**

| DIMES | | | | | |
|---|---|---|---|---|---|
| **MultiMM** | 0.725 | | | | |
| **MiChroM** | 0.406 | 0.382 | | | |
| **LoopSage** | 0.643 | 0.626 | 0.448 | | |
| **PHi-C2** | 0.593 | 0.574 | 0.373 | 0.610 | |
| | **DIMES** | **MultiMM** | **MiChroM** | **LoopSage** | **PHi-C2** |

**Table 4. Comparison of chromatin models generated based on ChIA-PET data. The table contains Spearman correlation coefficients calculated between pairs of distance matrices generated based on the models from five software packages (DIMES, MultiMM, MiChroM, LoopSage, and PHi-C2).**

| DIMES | | | | | |
|---|---|---|---|---|---|
| **MultiMM** | 0.275 | | | | |
| **MiChroM** | 0.248 | 0.413 | | | |
| **LoopSage** | 0.341 | 0.800 | 0.418 | | |
| **PHi-C2** | 0.365 | 0.776 | 0.413 | 0.751 | |
| | **DIMES** | **MultiMM** | **MiChroM** | **LoopSage** | **PHi-C2** |

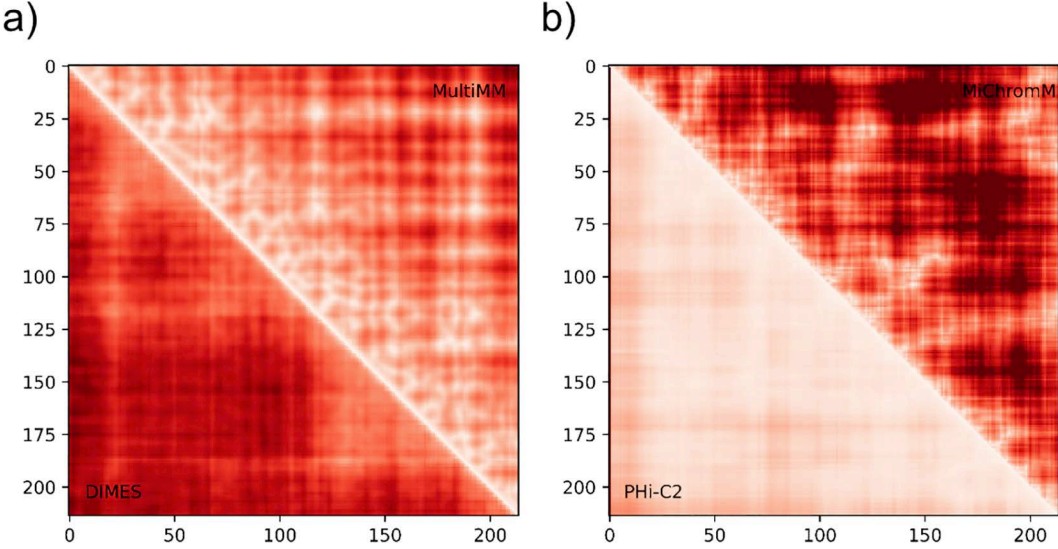

**Fig 6. Comparisons of heatmaps generated from Hi-C models with a) the highest correlation (DIMES vs MultiMM), and b) the lowest correlation (PHi-C2 vs MiChroM).** All four heatmaps correspond to the interpolated models (number of beads = 214) that were generated for the same region of interest (chr1:178.421.513–179.491.193) based on Hi-C data.

## Challenge of model validation

The second challenge we decided to address during the hackathon was the validation and interpretation of chromatin models. Such models aim to bridge theory and experiment, therefore, it is crucial to understand how experimental data underlies distances between genomic regions in the model and how close the model is to the real chromatin structure. This would advance the design of future experiments that aim to study the impact of the genome structure, i.e., proximity of various genomic regions (e.g., genes, promoters and enhancers) on gene expression and other cellular processes. During the hackathon, it was not easy to formulate the exact hypothesis and define the criteria for model validation. Firstly, chromatin models represent spatial distances between genomic regions, while experimental data can show contact probabilities (Hi-C), genomic interactions mediated by proteins involved in genome folding (ChIA-PET) or high-order genomic interactions (SPRITE). Those experimental methods complement each other, however, they provide different biological information. Furthermore, the difficulty of interpreting experimental data itself further impedes the challenge of model validation. Finally, there are currently no standard criteria or metrics to conduct such validation.

A variety of methods have been developed to validate 3C-based 3D chromatin inference algorithms [44]. Initially, in silico reference models were used to assess model behavior [34,35]. However, for optimal validation, models must be tested against real-world chromatin contact data. One common validation strategy for Hi-C-based models involves evaluating whether they accurately reproduce well-established chromatin features, such as chromosomal territories and compartment segregation. To quantitatively assess the accuracy of model reconstructions, image-based 3D measurements are often employed [30,36,37,40,41,87]. Among the most frequently used experimental datasets are 3D-FISH- and Oligopaint-based data from studies such as Beagrie et al. (2017) [55] and Bintu et al. (2018) [116].

A key challenge in validation of chromatin models is the availability of orthogonal datasets for the specific cell line used in modeling, which are not always accessible. In cases where fully orthogonal experimental data is unavailable, an alternative validation approach involves comparing the distance matrices derived from the model to the Hi-C contact map [35]. While this method is not orthogonal validation in the same sense as FISH-derived data, it still provides valuable insight into whether the chromatin structure inference method is functioning as intended. Specifically, it ensures that loci with a high number of Hi-C contacts correspond to short distances in the reconstructed distance matrix. This can be quantitatively assessed using correlation analysis or permutation tests.

Our project demonstrates that model validation is indeed a difficult task, even with expertise in both software and experimental data analysis. Here, we present our approach for model validation, in which we convert models into distance matrices, and then calculate Spearman correlation coefficients ($\rho$) between them to quantify model similarities. During the hackathon, we examined how models generated using Hi-C and ChIA-PET data correlate with three different biological data sets: Hi-C, ChIA-PET and SPRITE. To do this, we used the same chromatin models that we previously generated using Hi-C and ChIA-PET data for model comparison. We hypothesized that the distance matrices generated from Hi-C models would correlate more strongly with Hi-C data, while the matrices from ChIA-PET models would show a higher correlation with ChIA-PET data. To test our hypothesis, we calculated Spearman correlation coefficients between model matrices and the inverse of experimental contact frequency matrices. Our analysis revealed a significant heterogeneity in Spearman correlation coefficients across software packages, as well as the experimental datasets (see S1 Table and S2 Table). For instance, we observed the highest correlation between DIMES and Hi-C ($\rho = 0.66$) and the lowest correlation between MiChroM and SPRITE ($\rho = 0.19$). For models generated using Hi-C data, we observed a consistent pattern: they tended to correlate more strongly with Hi-C data compared to ChIA-PET or SPRITE data (Fig 7). Similar behavior was observed by using KL divergence as the comparison metric (Fig 8). Furthermore, we computed stratified correlations between simulated and experimental data. These results confirm that the models perform best at the resolution for which they were generated, while their accuracy decreases at finer resolutions (Fig 9).

The Hi-C models correlated well with Hi-C data, but interestingly, the ChIA-PET models also showed a stronger correlation with Hi-C data rather than ChIA-PET data. All methods were designed to work on data from chromosome

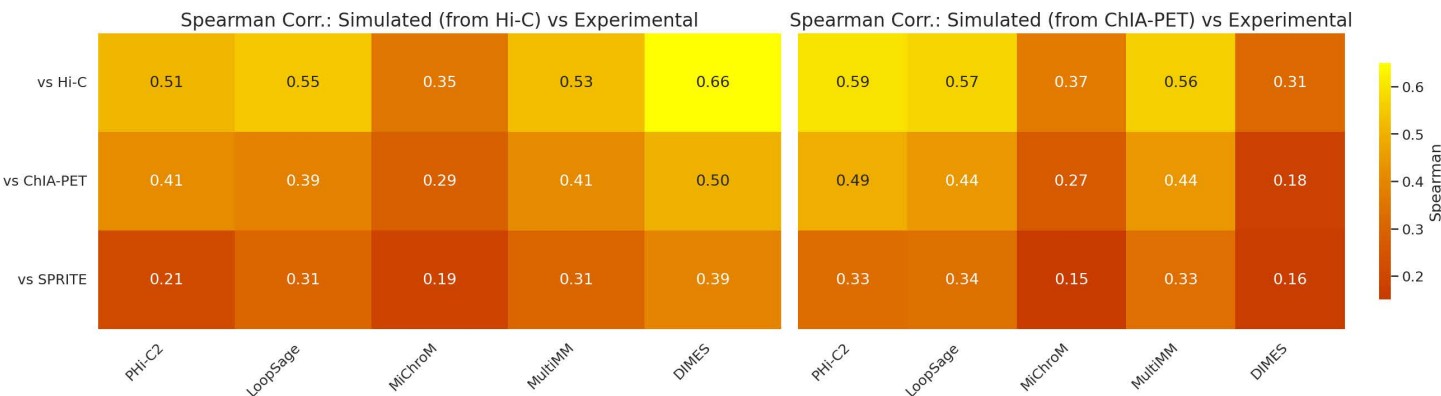

**Fig 7. Spearman correlation between simulated and experimental data.** The heatmaps display correlations for models generated from ChIA-PET and Hi-C data. Both types of models show stronger agreement with Hi-C experimental data, possibly due to the smoother structure of Hi-C-based models, which aligns more closely with experimental observations.

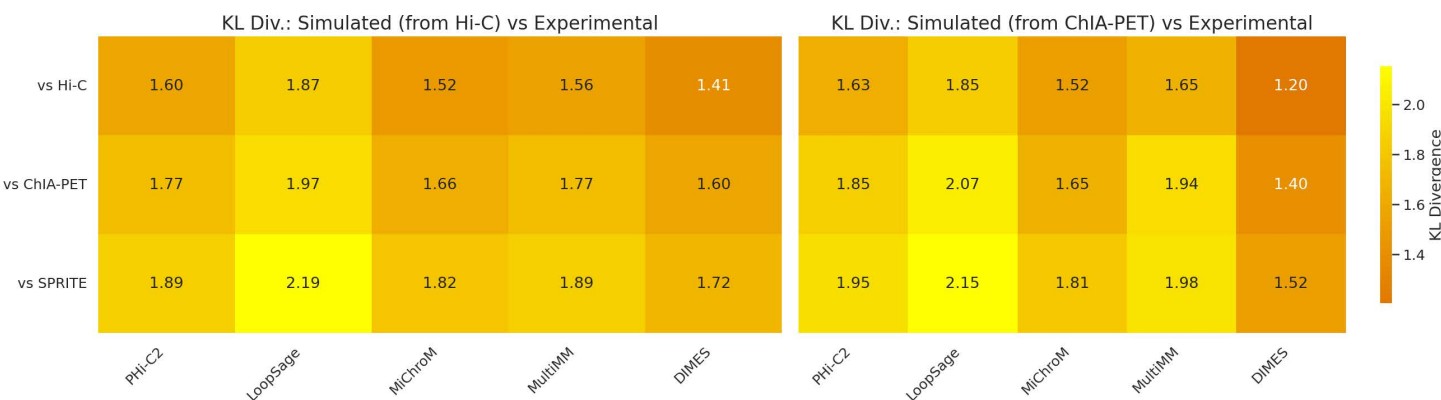

**Fig 8. KL divergence between simulated and experimental data.** The heatmaps show results for models generated from ChIA-PET and Hi-C data. Consistent with the Spearman correlation analysis, both model types exhibit better agreement with Hi-C experimental data.

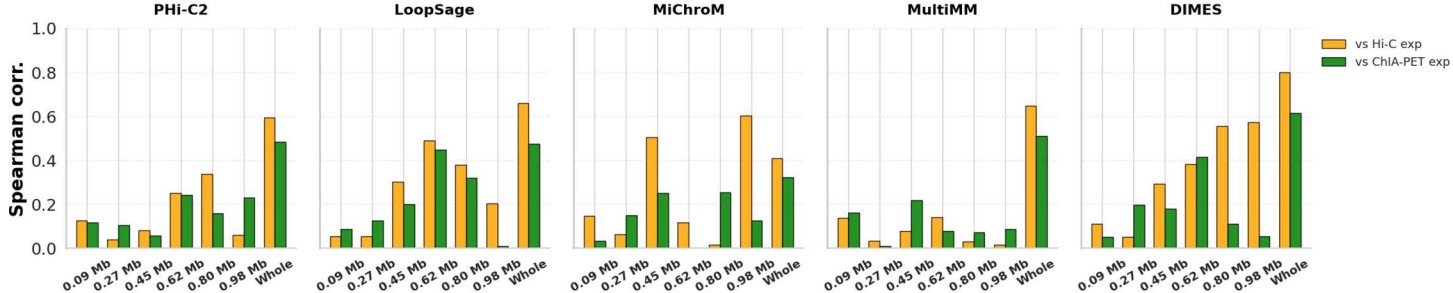

**Fig 9. Stratified Spearman correlations between simulated and experimental data.** Most models perform poorly in strata smaller than the resolution used to generate the final model. Typically, a simulation model shows good agreement only at the resolution it was designed to represent.

conformational capture techniques such as Hi-C, ChIA-PET or Hi-ChIP. Below, we describe for which particular data type each method was designed based on the information provided by the authors in the original articles. However, the authors of each method do not specify if the default parameters are adjusted to any particular data type or how to adjust them for other data types.

MiChroM and PHi-C2 were designed for Hi-C, therefore, the default parameters might work better for Hi-C than ChIA-PET. Similarly, HIPPS-DIMES was designed for Hi-C and/or imaging data (e.g., fluorescence in situ hybridization, FISH). On the other hand, LoopSage was originally designed to extract close chromatin regions from ChIA-PET data, but it can extract this information from Hi-C as well. One caveat, however, is that LoopSage is an ensemble-based model, and for fair comparison with other models, we generated a heatmap from a single representative structure. This simplification underscores that comparisons between ensemble-based approaches and deterministic top-down optimization models can be inherently difficult. MultiMM was designed for all 3C experiments: Hi-C, ChIA-PET, or Hi-ChIP.

In LoopSage, Hi-C data can be utilized in two ways: as input data or for validation. One can use Hi-C for both purposes or take an alternative approach—using ChIA-PET as input while employing Hi-C as an orthogonal dataset to estimate correlations between experimental and simulated heatmaps. This latter approach may offer a more robust validation method, as Hi-C heatmaps are population-averaged and generally smoother than ChIA-PET data. The increased smoothness can enhance correlation estimations, given that LoopSage heatmaps are also population-based and highly smoothed. Additionally, using orthogonal datasets improves the accuracy of validation, ensuring a more reliable assessment of the model's performance. Our results suggest that even though LoopSage was designed for ChIA-PET, the highest correlation of the output models is with Hi-C.

In summary, two primary factors may explain why models trained on ChIA-PET data exhibit a higher correlation with Hi-C maps. First, many of the methods analyzed in this study were specifically designed to process Hi-C data, rather than ChIA-PET, which may inherently bias their performance toward Hi-C datasets. Second, the structural properties of Hi-C interaction maps differ significantly from those of ChIA-PET maps. Hi-C experiments generate high-density interaction matrices, capturing a comprehensive range of chromatin contacts across the genome. In contrast, ChIA-PET selectively detects interactions associated with specific proteins, leading to a sparser, more discontinuous interaction map. This lower signal coverage across genomic regions may artificially inflate correlation estimates when comparing models trained on ChIA-PET to Hi-C maps, as the denser Hi-C-derived matrices provide a more continuous signal distribution.

## Discussion

Prior to the hackathon, we identified a lack of objective metrics to compare and validate 3D models of chromatin structure. It has been previously discussed that software performance, usability and interpretability are key aspects for studying genome folding [65,66,117,118], therefore, we set an aim for the 4D Nucleome Hackathon 2024 to address the challenges of chromatin model comparison and validation. Here we provide an overview of what is the current state of the 3D chromatin field. We start with a literature review of available software, and we show the variety of approaches for chromatin structure prediction, which leverage various experimental data types (e.g., Hi-C, ChIA-PET, ChIP-seq, imaging data or a combination thereof) and assume different modelling principles. We list example software packages and classify them by the scale of modeling, starting from the smallest scale - loops and TADs, through chromosomes to the whole genome (Tables 1 and 2). In addition, we describe several characteristics of such methods, including: 1) methods are usually designed to address specific aspects of chromatin biophysics, focusing on diverse biophysical problems and scales, which complicates software comparisons; 2) bioinformatic software frequently lacks long-term support and informative documentation [109]; 3) the complexity of chromatin folding necessitates expertise in biology, bioinformatics, and physics; 4) software for chromatin structure modeling requires objective metrics to quantify its efficiency.

**The first challenge** of our project was that a great number of software is not open-source, nor runnable without detailed technical knowledge and the field is lacking formal standardization. Our results indicate that it is indeed a difficult

problem, therefore, we emphasize the need for software accessibility and reproducibility, which would lower the entry barrier for young researchers to enter the field, thus enabling a quicker implementation of novel innovative ideas. In addition, there are no common guidelines for software development. We hypothesize that standardization and guidelines for software development, which are currently challenging to define, would have a positive long-term impact on the community.

**The second challenge** is the lack of objective criteria for model comparison and validation. Currently, there is no robust statistical metric to evaluate the quality of the experimental data. For that reason, state-of-the-art frameworks such as HiCRep [113] or GenomeDISCO [112] have been developed to assess the reproducibility of Hi-C that take into account the sensitivity-to-noise ratio and the unique spatial features of the data, including domain structures and distance-dependence. To address this challenge, we present a modular and scalable workflow for processing and comparing chromatin models, which includes a conversion of models into distance matrices and calculation of Spearman correlation coefficients between pairs of matrices that represent similarities between them. As a hackathon proof-of-principle, we compared models of one genomic region (a TAD of 1 Mb), obtained from five distinct software packages. We identified a big heterogeneity in the output models, which might be due to the variety of approaches and assumptions in the software. However, we acknowledge that this undertaking might be fraught with primary challenges, which were due to a limited time frame and resources. Nevertheless, we believe that our workflow provides a future reference for other initiatives that might be undertaken to develop criteria for chromatin model comparison and validation. For that purpose, we made our workflow publicly available on GitHub (https://github.com/SFGLab/Polymer_model_benchmark), and to ensure reproducibility, we provide scripts and virtual environment files to run on any Linux/GNU-based computing system.

Looking forward, it would be worthwhile to do a comprehensive study of all software for chromatin modeling, and especially to include 3D genomic methods incorporating artificial intelligence and single-cell technologies. Therefore, we plan to extend our joint effort to focus on those methodologies as well. It is crucial to examine how novel methodologies can advance the modeling itself, as well as the downstream analyses and model interpretation. Another potential avenue for improvement of chromatin modeling methods might be sought in the integration of 3D genomics data with multi-omic next-generation sequencing data to study the impact of genomic variation on the genome structure and function. To conclude, we identified and discussed the challenges that impede usability, reproducibility, and interpretability of the software for chromatin modeling, while emphasizing that chromatin modeling is crucial for biological and biomedical research.

## Materials and methods

### Software

During the hackathon, we used the following five software packages that incorporate different underlying methodologies based on various biophysical principles:

- LoopSage (https://github.com/SFGLab/LoopSage) [64],

- MiChroM (https://open-michrom.readthedocs.io/en/latest/OpenMiChroM.html) [75],

- DIMES (https://github.com/anyuzx/HIPPS-DIMES) [87],

- PHi-C2 (https://github.com/soyashinkai/PHi-C2) [85],

- MultiMM (https://github.com/SFGLab/MultiMM) [58].

Our workflow was implemented in Python (v3.12.2) using the NumPy (v1.26.4) and SciPy (v1.12.0) libraries.

### Data

Comparing various modeling techniques poses inherent challenges, necessitating the proposition of a methodologically straightforward approach for comparison. Initially, our focus was to evaluate the performance of these models within

small-scale topologically associated domain (TAD) regions and to assess their congruence with experimental data. In order to model the whole genome, a huge amount of computational resources, as well as time would be required. Due to the lack of those resources during the 4-day hackathon, a short genomic region of interest (chr1:178.421.513– 179.491.193) for the Tier 1 cell line GM12878 was chosen to generate the models. It is approximately 1 Mb long, and it represents a topologically-associated domain (TAD). We downloaded public data from the 4DNucleome Data Portal (https://data.4dnucleome.org/) [104,119] and ENCODE (https://www.encodeproject.org/) [120–122]. Chromatin models were generated based on the following in situ Hi-C data from the 4DN Data Portal: 4DNES4AABNEZ, 4DNESNMAAN97 and ENCODE: ENCSR968KAY, as well as ChIA-PET from ENCODE: ENCSR184YZV (CTCF), ENCSR764VXA (SMCA1). For model validation, we used SPRITE data from 4DN Data Portal: 4DNESI1U7ZW9.

## Supporting information

**S1 Table. Comparison of chromatin models generated based on Hi-C with experimental data.**
(XLSX)

**S2 Table. Comparison of chromatin models generated based on ChIA-PET with experimental data.**
(XLSX)

## Acknowledgments

We thank The 4D Nucleome Consortium for organizing and sponsoring the 4D Nucleome Hackathon 2024, and The University of Washington for hosting the event. Part of the high-performance computations were performed thanks to the Laboratory of Bioinformatics and Computational Genomics, Faculty of Mathematics and Information Science, Warsaw University of Technology using Artificial Intelligence HPC platform financed by Polish Ministry of Science and Higher Education (decision no. 7054/IA/SP/2020 of 2020-08-28).

## Author contributions

**Conceptualization:** Jędrzej Kubica, Krzysztof H. Banecki , Michał Kadlof, Ben Busby, Dariusz Plewczynski.

**Data curation:** Jędrzej Kubica, Sevastianos Korsak, Dvir Schirman, Anurupa Devi Yadavalli, Ariana Brenner Clerkin, David Kouřil.

**Formal analysis:** Jędrzej Kubica, Sevastianos Korsak, Dvir Schirman, Anurupa Devi Yadavalli, Ariana Brenner Clerkin, David Kouřil.

**Funding acquisition:** Dariusz Plewczynski.

**Investigation:** Jędrzej Kubica, Sevastianos Korsak, Dvir Schirman, Anurupa Devi Yadavalli, Ariana Brenner Clerkin, David Kouřil.

**Methodology:** Jędrzej Kubica, Sevastianos Korsak, Dvir Schirman, Anurupa Devi Yadavalli, Ariana Brenner Clerkin, David Kouřil.

**Project administration:** Jędrzej Kubica.

**Resources:** Dariusz Plewczynski.

**Software:** Jędrzej Kubica, Sevastianos Korsak, Dvir Schirman, Anurupa Devi Yadavalli, Ariana Brenner Clerkin, David Kouřil.

**Supervision:** Michał Kadlof, Ben Busby, Dariusz Plewczynski.

**Validation:** Jędrzej Kubica.

**Visualization:** Sevastianos Korsak, Krzysztof H. Banecki , Dvir Schirman, Anurupa Devi Yadavalli, Ariana Brenner Clerkin, David Kouřil, Michał Kadlof.

**Writing – original draft:** Jędrzej Kubica, Sevastianos Korsak, Krzysztof H. Banecki , Michał Kadlof.

**Writing – review & editing:** Jędrzej Kubica, Sevastianos Korsak, Krzysztof H. Banecki , Dvir Schirman, Anurupa Devi Yadavalli, Ariana Brenner Clerkin, David Kouřil, Michał Kadlof, Ben Busby, Dariusz Plewczynski.

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
