## [Decision Letter · Decision Letter 0]

26 Dec 2024

PCOMPBIOL-D-24-01707

The challenge of chromatin model comparison and validation: a project from the first international 4D Nucleome Hackathon

PLOS Computational Biology

Dear Dr. Plewczynski,

Thank you for submitting your manuscript to PLOS Computational Biology. After careful consideration, we feel that it has merit but does not fully meet PLOS Computational Biology's publication criteria as it currently stands. Therefore, we invite you to submit a revised version of the manuscript that addresses the points raised during the review process.

Please submit your revised manuscript within 60 days Feb 25 2025 11:59PM. If you will need more time than this to complete your revisions, please reply to this message or contact the journal office at ploscompbiol@plos.org. Please include the following items when submitting your revised manuscript:

We look forward to receiving your revised manuscript.

Kind regards,

Jie Liu

Academic Editor

PLOS Computational Biology

Jian Ma

Section Editor

PLOS Computational Biology

**Journal Requirements:**

3) We notice that your supplementary Tables are included in the manuscript file. Please remove them and upload them with the file type 'Supporting Information'. Please ensure that each Supporting Information file has a legend listed in the manuscript after the references list.

4) Please amend your detailed Financial Disclosure statement. This is published with the article. It must therefore be completed in full sentences and contain the exact wording you wish to be published.

5) Please ensure that the funders and grant numbers match between the Financial Disclosure field and the Funding Information tab in your submission form. Note that the funders must be provided in the same order in both places as well. Currently, the order of the grants is different in both places.

Please indicate by return email the full and correct funding information for your study and confirm the order in which funding contributions should appear. Please be sure to indicate whether the funders played any role in the study design, data collection and analysis, decision to publish, or preparation of the manuscript.

**Reviewers' comments:**

Reviewer's Responses to Questions

Reviewer #1: Kubica et al. reported the findings of their analyses from the first 4D Nucleosome Hackathon, 2024. The authors designed workflows to benchmark five different computational methods for predicting 3D chromatin models from experimental Hi-C and ChIA-PET data sets. The authors compared the predictions at a specific 1Mb locus (chr1:178,421,513–179,491,193) in the GM12878 cell line. Since all the methods predicted 3D chromatin at different resolutions, the authors selected the predictions corresponding to 214 beads from the 1Mb window from all the methods. Here, each bead corresponds to approximately 5,000 base-pairs. The authors found the models trained using Hi-C, generated different predictions compared to the models trained using ChIA-PET. Furthermore, they compared the predictions across models trained using the same experimental data and stated that the predictions are variable across the five methods. Likewise, the authors compared the model predictions to experimental data from Hi-C, ChIA-PET, and SPRITE using Spearman correlation, and found that the models trained using a specific experimental data do not necessarily correlate with the given experimental data.

These analyses highlight the current limitations in existing methods for predicting 3D chromatin models. They also point out the need to understand the differences across the various experimental data sets as there is variability in the contact maps generated using different experimental protocols. Due to the lack of sufficient time for the hackathon, the benchmarking was done on a small set of methods at a single genomic locus. In future, it would be useful if such analyses were performed on a larger scale.

Comments:

1. How were the five methods selected for benchmarking? The authors report these five methods as being user-friendly. Were these the only user-friendly methods or were there additional criteria for selecting these methods? What were the significant difficulties faced with each software in Table 1?

2. What were the default resolutions for each of the five methods before selecting 214 beads?

3. In Fig. 6b, are the models trained on ChIA-PET and not Hi-C?

4. Can the Spearman correlations be split by distance between the genome contacts? For example, all contacts within 250kb, 250-500kb, etc..

5. The authors point out that the models trained using ChIA-PET have the highest correlation with Hi-C instead of ChIA-PET. Is it possible that these methods were originally designed for Hi-C, so the default parameters work best for predicting Hi-C contacts?

Reviewer #2: Kubica et al. presented a hackathon project aimed at comparing and validating 3D chromatin models. 3D model comparisons and validations are of interest to the field, though several major issues need to be addressed beyond the current analysis results.

Major issues:

1. A more clearly stated focus could improve the manuscript. Right now, the title and the first part look like a review or benchmark of different scales of interactions and various prediction tasks, but the results are mostly focused on inferring the 3D distance based on Hi-C or ChIA-PET.

2. Only a general Spearman correlation is used to compare 3D distance results. As motivated in the first part of the manuscript, it would be interesting to assess 3D interactions in different scales.

3. Ensemble methods are supposed to mitigate the noise of the inference, whereas, in comparison, different methods have different choices of ensembling, which could drive the difference in comparisons. Thus, ensembling information need to be further investigated or controlled for in the comparison.

4. Another major issue is validation. Although the authors acknowledge it is hard to validate, it will be more informative if orthogonal information (e.g., image-based 3D distance measurements) can be used to compare with 3D models based on Hi-C and ChIA-PET. It is not clear to me how and why Hi-C-based distance is used to evaluate Hi-C-based 3D models’ distance.

5. How does inference based on micro-C compare to inference based on Hi-C and ChIA-PET?

6. Currently, the comparison result is mostly limited to one cell line. Although the time is limited for a hackathon, it would be more conclusive if more cell lines were included.

Minor issues:

1. The color scheme is lacking, and the legends are the same for Figures 5 and 6.

Reviewer #3: The authors report the challenge and results of one of the projects for 4D Nucleome Hackathon, aimed at benchmarking computational models of chromatin structure modeling. The authors highlight the complexity and hierarchical organization of chromatin, as well as the variety of modeling approaches and scales that contribute to the difficulty of model comparison and validation. They propose a workflow that uses Spearman correlations to quantify similarity between models and between models and experimental data. Their work underscores the importance of standardized guidelines and improved software practices to advance the field of 3D genome modeling. I have a few comments:

1. In Introduction, ChIA-PET and Hi-ChIP experiments produce contact matrices and prove ef�cacious in TAD-scale modeling. Shouldn't ChIA-PET and Hi-ChIP are used mainly for detecting loops?

2. Related to point 1, ChIA-PET is used for identifying looping that involve specific proteins. The authors should specify which ChIA-PET data they are using (in this case I believe it's CTCF and SMC1A)

3. In the introduction the author briefly touched upon single cell Hi-C, I think it's worth mentioning other single cell multi-omics with chromatin interactions included such as sn-m3c-seq, HiRES, GAGE-seq as they facilitate the training of predictive modeling at single cell resolutions. Related to this, SPRITE, ChIA-Drop and other potential multi-way interaction profiling method could also motivate new models being created.

4. When discussing existing works, I think it's also worth to discuss existing evaluation metrics for these models as this paper is mainly about evaluation, discussing previous work on this part is crucial. For example, the Distance-Stratified Correlation Coefficients, softwares like GenomDISCO, HiCRep etc.

5. I suggest for Table 1, split it into several tables based on if it's predictive modeling or polymer modeling etc. currently it's a long table and a mixture of different models.

**Have the authors made all data and (if applicable) computational code underlying the findings in their manuscript fully available?**

Reviewer #1: Yes

Reviewer #2: Yes

Reviewer #3: None

PLOS authors have the option to publish the peer review history of their article (what does this mean? ). If published, this will include your full peer review and any attached files.

**Do you want your identity to be public for this peer review?** For information about this choice, including consent withdrawal, please see our Privacy Policy .

Reviewer #1: No

Reviewer #2: No

Reviewer #3: No

**Figure resubmission:**
---

## [Decision Letter · Decision Letter 1]

14 Apr 2025

PCOMPBIOL-D-24-01707R1

The challenge of chromatin model comparison and validation: a project from the first international 4D Nucleome Hackathon

PLOS Computational Biology

Dear Dr. Plewczynski,

Thank you for submitting your manuscript to PLOS Computational Biology. After careful consideration, we feel that it has merit but does not fully meet PLOS Computational Biology's publication criteria as it currently stands. Therefore, we invite you to submit a revised version of the manuscript that addresses the points raised during the review process.

Please submit your revised manuscript within 60 days Jun 14 2025 11:59PM. If you will need more time than this to complete your revisions, please reply to this message or contact the journal office at ploscompbiol@plos.org. Please include the following items when submitting your revised manuscript:

We look forward to receiving your revised manuscript.

Kind regards,

Jie Liu

Academic Editor

PLOS Computational Biology

Jian Ma

Section Editor

PLOS Computational Biology

**Journal Requirements:**

1) We note that your Figures files are duplicated on your submission. Please remove any unnecessary or old files from your revision, and make sure that only those relevant to the current version of the manuscript are included.

2) We notice that your supplementary Tables are included in the manuscript file. Please remove them from the main file of the manuscript . Please note that supporting information should be uploaded separately with the file type 'Supporting Information'.

**Reviewers' comments:**

Reviewer's Responses to Questions

Reviewer #1: The authors have addressed most of my comments. However, it is not clear whether they calculated Spearman correlations split by distance. In response to comments 1.4 and 2.2 they stated that they calculated different metrics based on distance stratifications but I was unable to find the related section and figures in the manuscript. Could the authors highlight the relevant sections? Additionally, calculating these metrics based on distance stratification is important because it would highlight which models are more accurate in predicting distal enhancers which are key elements of gene regulation.

Reviewer #2: Thank you for your efforts in trying to address many of the points raised. I have several follow-up concerns on some of the points:

Regarding Reviewer 2, point 4, it is still concerning that the evaluation labels overlap with the input. This may weaken both the study design and the strength of the conclusions. The authors opted to retain the original validation approach, citing that “some methods do incorporate Hi-C matrices as part of their validation strategies in the absence of available orthogonal image-based information (e.g., Carstens et al., 2016).” However, this rationale seems problematic. In this setup, a model that simply reproduces the input Hi-C distances could achieve seemingly perfect performance, which would not be informative. Thus, orthogonal ways to evaluate seems key to draw informative conclusions.

Additionally, several critical analyses mentioned in the response letter do not appear to be included in the manuscript. Specifically, I was unable to locate the distance-based comparisons or the KL-divergence analyses brought up in Reviewer 1, point 4, and Reviewer 2, point 2.

Reviewer #3: The authors have addressed all my previous questions, I have no further comments

**Have the authors made all data and (if applicable) computational code underlying the findings in their manuscript fully available?**

Reviewer #1: Yes

Reviewer #2: Yes

Reviewer #3: None

PLOS authors have the option to publish the peer review history of their article (what does this mean? ). If published, this will include your full peer review and any attached files.

**Do you want your identity to be public for this peer review?** For information about this choice, including consent withdrawal, please see our Privacy Policy .

Reviewer #1: No

Reviewer #2: No

Reviewer #3: No

**Figure resubmission:**
---

## [Decision Letter · Decision Letter 2]

22 Jul 2025

Dear Professor Plewczynski,

We are pleased to inform you that your manuscript 'The challenge of chromatin model comparison and validation: a project from the first international 4D Nucleome Hackathon' has been provisionally accepted for publication in PLOS Computational Biology.

Best regards,

Jie Liu

Academic Editor

PLOS Computational Biology

Jian Ma

Section Editor

PLOS Computational Biology

There is a fourth reviewer who provided minor revision suggestion. "The authors have made substantial and thoughtful revisions in response to the reviewer feedback. The manuscript is nearly ready for acceptance. However, given the hackathon constraints and the focus on a single genomic region, the generalizability of the conclusions to other genomic contexts (e.g., compartment boundaries, heterochromatin) remains uncertain. I recommend the authors add a brief acknowledgment of the limited genomic scope of their analysis in the discussion section to clarify the generalizability of their findings." Please consider adopting it, but it is not required.

Reviewer's Responses to Questions

**Comments to the Authors:**

Reviewer #1: The authors have addressed all my comments.

Reviewer #2: The authors have addressed all my questions, I have no further comments.

Reviewer #3: The authors have addressed my concerns, I have no further comments

Reviewer #4: The authors have made substantial and thoughtful revisions in response to the reviewer feedback. The manuscript is nearly ready for acceptance. However, given the hackathon constraints and the focus on a single genomic region, the generalizability of the conclusions to other genomic contexts (e.g., compartment boundaries, heterochromatin) remains uncertain. I recommend the authors add a brief acknowledgment of the limited genomic scope of their analysis in the discussion section to clarify the generalizability of their findings.

**Have the authors made all data and (if applicable) computational code underlying the findings in their manuscript fully available?**

Reviewer #1: Yes

Reviewer #2: None

Reviewer #3: None

Reviewer #4: Yes

PLOS authors have the option to publish the peer review history of their article (what does this mean? ). If published, this will include your full peer review and any attached files.

**Do you want your identity to be public for this peer review?** For information about this choice, including consent withdrawal, please see our Privacy Policy .

Reviewer #1: No

Reviewer #2: No

Reviewer #3: No

Reviewer #4: **Yes: ** Yuanhao Huang

---

## [Editor Report · Acceptance letter]

PCOMPBIOL-D-24-01707R2

The challenge of chromatin model comparison and validation: a project from the first international 4D Nucleome Hackathon

Dear Dr Plewczynski,

I am pleased to inform you that your manuscript has been formally accepted for publication in PLOS Computational Biology. Your manuscript is now with our production department and you will be notified of the publication date in due course.

With kind regards,

Zsofia Freund
